# A Compact Modular 5 GW Pulse PFN-Marx Generator for Driving HPM Source

**Haoran Zhang [1,2], Ting Shu [1,2], Shifei Liu [1,2], Zicheng Zhang [1,2,*], Lili Song [1,2,*] and Heng Zhang [1,2]**

1    College of Advanced Interdisciplinary Studies, National University of Defense Technology,
     Changsha 410073, China; hrzhang@nudt.edu.cn (H.Z.); ting.shu@nudt.edu.cn (T.S.);
     liushifei16@nudt.edu.cn (S.L.); zhnudt@126.com (H.Z.)
2    State key Laboratory of Pulsed Power Laser Technology, Changsha 410073, China
*    Correspondence: zczhang@nudt.edu.cn (Z.Z.); phoezy@126.com (L.S.)

**Abstract:** A compact and modular pulse forming network (PFN)-Marx generator with output parameters of 5 GW, 500 kV, and 30 Hz repetition is designed and constructed to produce intense electron beams for the purpose of high-power microwave (HPM) generation in the paper. The PFN-Marx is composed by 22 stages of PFN modules, and each module is formed by three mica capacitors (6 nF/50 kV) connected in parallel. Benefiting from the utilization of mica capacitors with high energy density and a mini-trigger source integrated into the magnetic transformer and the magnetic switch, the compactness of the PFN-Marx system is improved significantly. The structure of the PFN module, the gas switch unit, and the connection between PFN modules and switches are well designed for modular realization. Experimental results show that this generator can deliver electrical pulses with the pulse width of 100 ns and amplitude of 500 kV on a 59-ohm water load at a repetition rate of 30 Hz in burst mode. The PFN-Marx generator is fitted into a cuboid stainless steel case with the length of 80 cm. The ratio of storage energy to volume and the ratio of power to weight of the PFN-Marx generator are calculated to be 6.5 J/L and 90 MW/kg, respectively. Furthermore, utilizing the generator to drive the transit time oscillator (TTO) at a voltage level of 450 kV, a 100 MW microwave pulse with the pulse width of 20 ns is generated.

**Keywords:** PFN-Marx; compact; modular; trigger source; gas switch; mica capacitor



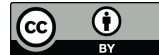

## 1. Introduction

Intense electron beams have a wide range of applications in science research and industry fields, such as in the high-power microwave (HPM) and flash X-ray [1–3]. Producing high voltage (with several hundred kilovolts), rectangular pulses are one of the crucial parts. Recently, with the development of the energy storage technology, pulsed power technology has been developed in the direction of compactness, miniaturization, and light weight. Because the pulse forming network (PFN)-Marx generator has the natural advantage of the integration of pulse modulation and pulse voltage accumulation, it has been developed rapidly in recent years and is regarded as one of the most promising types for compactness, modularization, light weight, and miniaturization realization [4–9]. In this paper, two key points, including the compactness and modularization, of the PFN-Marx generator are discussed in detail.

Firstly, the literature on the research of the Marx generator over the past few decades has been reviewed. Institutes around the world, including the French-German Research Institute of Saint-Louis (ISL), Texas Technology University (TTU), Commissariat à l'Energie Atomique et aux Energies Alternatives (CEAE), Applied Physical Electronics L. C. (APELC), etc., have developed a series of compact Marx generators with different circuit topologies and the typical parameters of the generators are listed in Table 1 [10–15]. Generally, the ratio of storage energy to volume (E/V) and the ratio of power to weight (P/W) are calculated to describe the compactness of the generator. According to the limited parameters of the

generator in the literature, the E/V and P/W are calculated to several J/L and dozens of MW/kg, respectively. Actually, a variety of methods, including structure optimization, novel topology circuits, large energy density capacitors, etc., have been studied by researchers to realize the compact design. However, the physical limits of insulation are roughly the same in spite of the circuit topologies and structures. The effective way to improve the compactness of the generator is by using the high energy density capacitors. Generally, the ceramic capacitors and the film capacitors are more popular in Marx generator applications. The ceramic capacitors have the advantages of small size and low self-inductance, and the film capacitors have the advantages of large capacitance and high withstand voltage. In terms of compact Marx generators, ceramic capacitors are used more often because of their small size. However, the energy density of ceramic capacitors is low because it is difficult to balance the large capacitance, high withstand voltage, and small volume. Recently, mica capacitors have gradually drawn researchers' attention because of their good electric characteristics [11]. Mica capacitors have the advantages of large capacitance, high withstand voltage, and small size, which make them a promising alternative to ceramic capacitors in the compact Marx generators.

**Table 1.** The parameters of the typical compact pulse forming network (PFN)-Marx generators.

| Research Institute | Output Power | Output Voltage | Pulse Width | E/V | P/W |
|---|---|---|---|---|---|
| ISL | 2.3 GW | 342 kV | 60 ns | 2 J/L | 70 MW/kg |
| TTU | 2.2 GW | 210 kV | 200 ns | 9 J/L | - |
| CEAE | 1.6 GW | 400 kV | 85 ns | 2 J/L | 36 MW/kg |
| APELC | 5 GW | 350 kV | 200 ns | 2 J/L | - |

Secondly, the modular design of the PFN-Marx generator generally contains two meanings. One is the modular implementation of the PFN-Marx body. Obviously, a variety of PFN-Marx generators with different structures have been designed. However, once the generator is assembled, the PFN module is difficult to replace. The modular design requires that the PFN module can be taken out and replaced from the system individually and therefore the PFN-Marx generator can be convenient for maintenance and parameter adjustment. In this case, the optimization of the PFN module design and assembling method becomes necessary. On the other hand, the PFN-Marx generator system includes many subsystem modules by their functions, which are the primary energy module, PFN-Marx body, trigger source, and vacuum diode. In order to realize a compact PFN-Marx generator system, the design of the subsystem modules should be optimized as well.

In this paper, mica capacitors with large energy density are assembled in the PFN-Marx generator to improve its compactness. The PFN module and the integrated gas gap switch are well designed for modular realization. In Section 2, the design and the assembly of the PFN module are elaborated in detail and the subsystem module of the PFN-Marx system is introduced in Section 3, followed by experimental results and a summary in Sections 4 and 5, respectively.

## 2. Modular PFN-Marx Design

### 2.1. Overall Description

The PFN-Marx generator is designed for driving a HPM source. In order to make the generator have a wider range of application, the peak power (P) and the load impedance (R) are designed as 5 GW and 50–60 Ω, respectively.

Figure 1 shows the schematic circuit of the PFN-Marx. The type-E is applied in this generator, because the equal L-C sections are easy for modular realization and fabrication in practice. The established parameters of the PFN-Marx are listed in Table 2. The number of the node capacitor n and the number of the PFN module m can be calculated to 3 and 22, respectively, by Equations (1)–(3).

$$\tau = 2n\sqrt{LC} \tag{1}$$

$$R = m\sqrt{\frac{L}{C}} \tag{2}$$

$$V_{output} = \frac{mV_{ch}}{2} \tag{3}$$

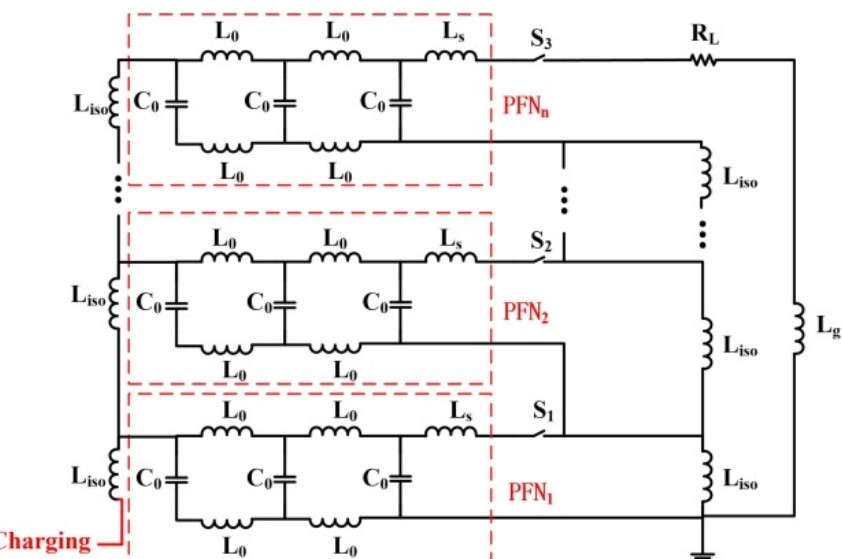

**Figure 1.** Schematic circuit of the PFN-Marx.

**Table 2.** PFN-Marx generator specifications.

| Symbol | Description | Value |
|--------|-------------|-------|
| P | Peak power | 5 GW |
| $V_{load}$ | Load voltage | 500 kV |
| R | Load impedance | 50–60 Ω |
| T | Pulse width | 90–100 ns |
| $C_0$ | Node capacitor | 6 nF |
| $L_0$ | Node inductor | 15 nH |
| $V_{ch}$ | Charging voltage | 50 kV |
| Rep | Repetition rate | 30 Hz |

In order to realize the repetition operation of the generator, the inductive isolation of the charging is used and the optimized isolation inductor $L_{iso}$ is calculated to 20 μH. The inductance of the switch and the switch leading wire $L_s$ is set as 45 nH. The simulation result is shown in Figure 2. The load voltage waveform characteristics include the rise-time of about 20 ns, pulse width of about 90 ns, flat-top of about 50 ns, and amplitude of about 530 kV with the charging voltage of 50 kV.

A series of methods are used for the realization of the modular PFN-Marx body, including the PFN module design, the connection method between the PFN module and the electrode of the switch, and the common switch housing. The key components of the modular PFN-Marx generator are illustrated as follows.

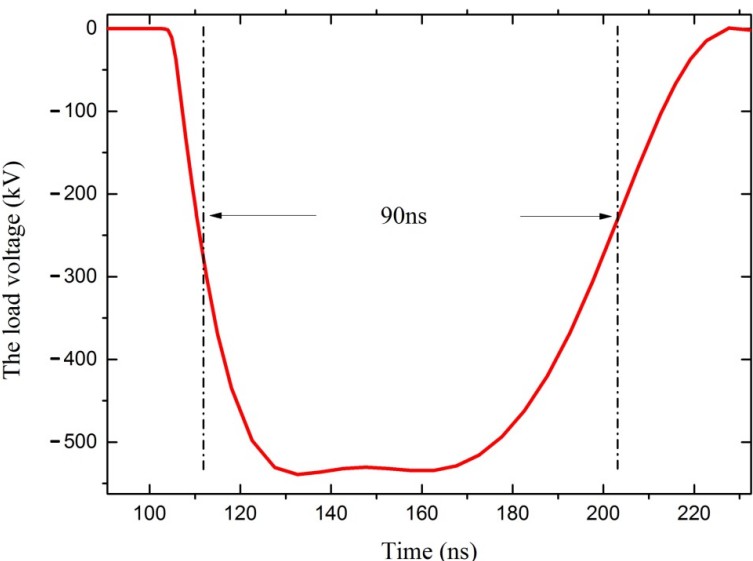

**Figure 2.** The load voltage waveform in simulation.

### 2.2. PFN Module

The assembly diagram of the PFN module is shown in Figure 3. Each PFN module consists of three 6000 pF ($22 \times 40 \times 95$ mm³) mica capacitors connected in parallel by means of copper strips, and the copper strips also serve as the node inductors. The mica capacitors are inserted into a nylon case with external dimensions of 178 mm $\times$ 150 mm $\times$ 30 mm. The energy density of the mica capacitor is nearly 90 J/L, which is over three times that of the commercial ceramic capacitor with 2000 pF ($\varphi$60 mm $\times$ 32 mm). The selection of the mica capacitor with large energy density sets a good foundation for the compact design. Each mica capacitor assembly was tested under the DC voltage of 50 kV with a time duration of 1 min before usage. Then, the connected mica capacitors were inserted into a nylon box to ensure the insulation distance between the adjacent modules and to provide mechanical support. Finally, the key point of the modular realization is the connection between the PFN module and the switch. Here, the socket joint mode is adopted in which two plug electrodes are welded with the PFN via two high-voltage wires. In this way, the PFN module can be taken out individually from the PFN-Marx body and the structure of the PFN-Marx remains unchanged.

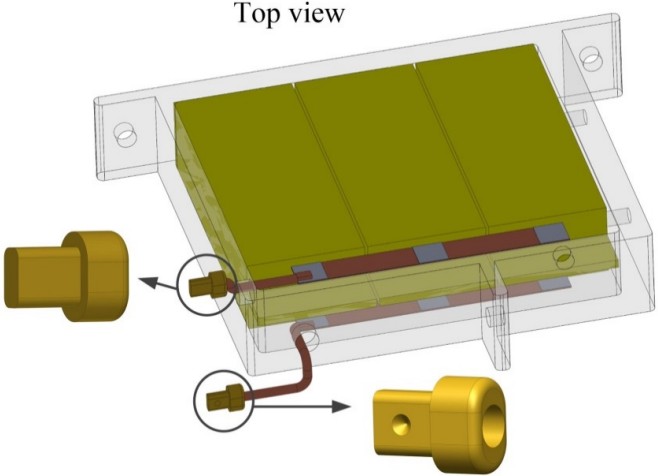

**Figure 3.** The assembly diagram of the PFN module.

### 2.3. Gas Gap Switches

Switches, as energy-shifting elements in the pulse power system, play an important role in the pulse compression and power enhancement. Generally, the generation of high voltage of the PFN-Marx generator is that the PFN modules are erected via the Marx-type method. Therefore, the number of switches of the PFN-Marx generator is the same as the number of the PFN modules. For a PFN-Marx generator, the operating characteristics of switches are related to the wave erection. So, the switches are designed specifically to ensure the performance and at the same time to realize the modular function of the PFN-Marx generator. In practice, all of the switch electrodes of the generator are assembled in a nylon housing with external dimensions of 695 mm × 85 mm × 60 mm, as shown in Figure 4. Actually, there are two main advantages of the integrated switch design: one is that the gas pressure of the switch can be adjusted independently, so that the operation voltage of the switch can be adjusted in a wide range; the other advantage is that the ultraviolet light, which is generated when the first few stages of the switches close, facilitates the switching of the later ones. In this way, the erected time of the PFN-Marx can be decreased.

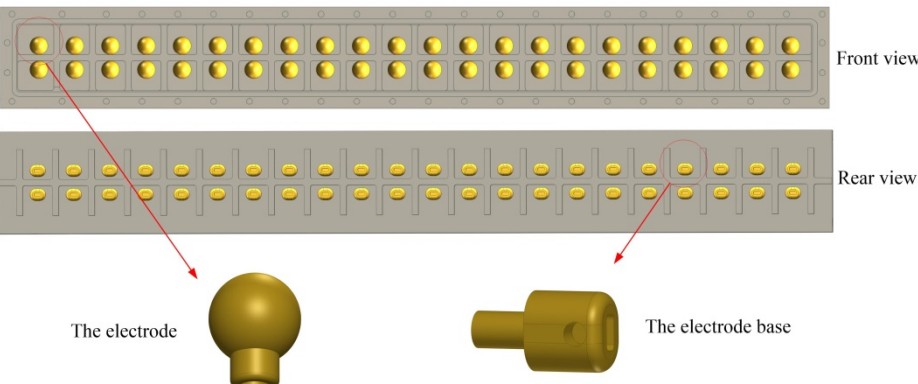

**Figure 4.** The assembly diagram of the gas gap switches.

The electrode of the switch has a spherical geometry with the material of copper. The diameter of the electrode is 15 mm and the gap distance is 10 mm. These spherical electrodes are assembled with the electrode bases, as shown in Figure 4. A rectangular countersink hole is milled at the end of the electrode base to assemble the plug electrode, which is shown in Figure 3. The plug electrode and the electrode base are secured by a metal screw. Therefore, the switch of the generator is an independent assembly and the PFN module can realize the quick dismantling function.

In fact, in order to improve the operation stability of the generator, the switch is normally triggered by a high-voltage electric pulse. The switches of the first two stages are a three-electrode configuration constructed by a 2 mm stainless steel needle into a 15 mm brass sphere electrode, and Figure 5 shows the schematic diagram of the three-electrode switch. The gap length between triggered electrode and main electrode and between two main electrodes is 3 mm and 10 mm, respectively. The dielectric material is assembled in the electrode to insulate the trigger electrode. The lifetime of the single triggered switch was tested in the experiment. The insulation gas is $N_2$ mixed with 15% $SF_6$ and the gas pressure in the switch house is 80 kPa. The switch was tested in 10 Hz repetition mode with an operation voltage of 50 kV over 10,000 shots. Figure 6 shows the photograph of the triggered switch after 100,000 shots. This work indicates that the gas gap switch in Figure 4 would have a long lifetime when it periodically changes the working gas.

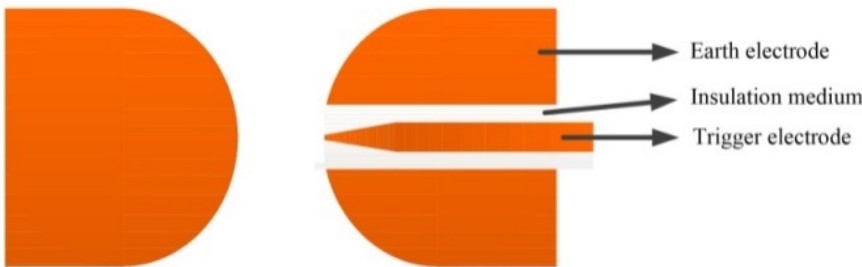

**Figure 5.** Schematic diagram of the trigger switch.

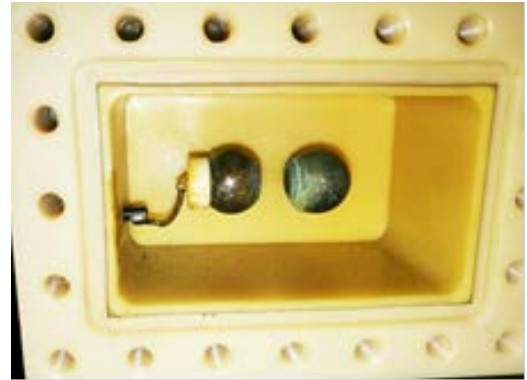

**Figure 6.** Photograph of the trigger switch.

### 2.4. The Modular PFN-Marx Generator Assembly

Twenty-two PFN modules are constructed in the form of a drawer-type geometry, which can minimize circuit inductance as well as realize the modular function, as shown in Figure 7. There are four components of the PFN-Marx body, including three support rods, PFN modules, and a switch assembly. Firstly, the PFN modules are assembled on nylon rods with nylon screws. Secondly, the switch assembly, the nylon rods, and the PFN modules are assembled on the nylon plate. Finally, the plug electrodes are inserted into the electrode bases of the switch and assembled with metal screws. In this way, the PFN module can be taken out individually and the PFN-Marx structure is kept unchanged. Therefore, the convenience of the PFN-Marx generator's maintenance can be improved significantly.

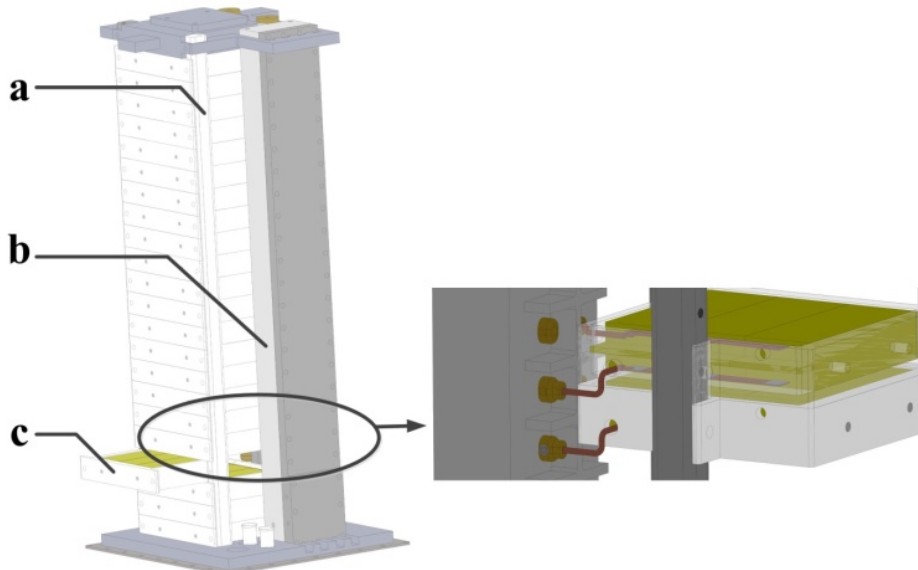

**Figure 7.** Assembly diagram of the PFN-Marx (a—support rod, b—gas switch, c—PFN module).

## 3. Experimental Setup

### 3.1. Primary Energy Subsystem

A primary energy subsystem was designed and established for repetition operation of the PFN-Marx generator, and the equivalent circuit is shown in Figure 8. At the beginning of the generator operation, the intermediate energy storage capacitor $C_f$ and the primary capacitor of the pulse transformer $C_p$ are charged by an AC transformer $TF_1$, and the maximum charging voltage is 2800 V. The mica capacitors in the PFN-Marx generator are charged by an air-core pulse transformer TF and the maximum charging voltage $V_{ch}$ is up to 50 kV. The charging voltage waveform is shown in Figure 9.

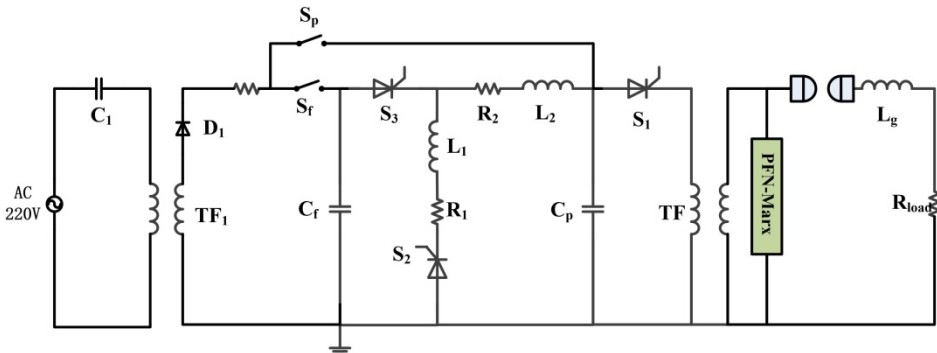

**Figure 8.** Schematic circuit of the primary energy subsystem.

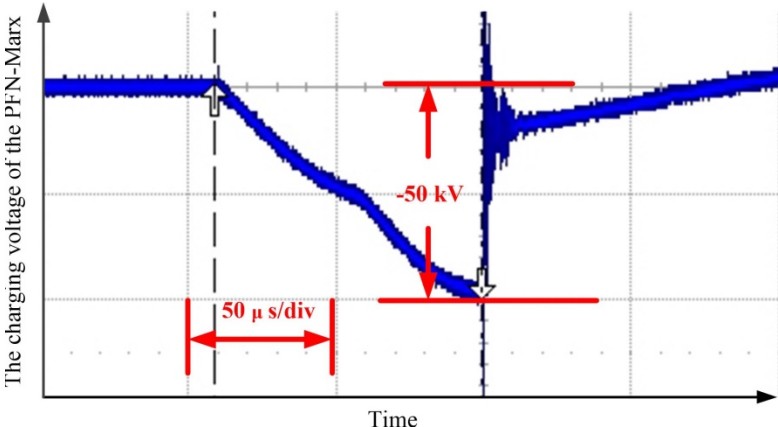

**Figure 9.** The charging voltage waveform of the PFN-Marx generator.

The working process of the system in the case of repetition mode is described as follows. The operation sequence of the thyristors is shown in Figure 10. Firstly, the $C_f$ and $C_p$ are charged to $U_f$ and $U_0$, respectively. Next, when the thyristor $S_1$ is closed at time $T_1$, the $C_p$ will discharge to the PFN-Marx through the TF. After that, the $C_p$ has the negative residual voltage $U_{rec}$ because of the principle of the unidirectional current of the thyristor. Then, the thyristor $S_2$ is closed at time $T_2$, and the voltage polarity of the $C_p$ is reversed via the loop of $C_p$-$S_2$-$R_1$-$L_1$-$R_2$-$L_2$-$C_p$. Finally, the thyristor $S_3$ is closed at time $T_3$, the $C_p$ is charged to $U_0$ again, and the system waits for the next operation command. So, the key point of the repetition operation is that the control of the on time of the thyristors and the pulse number is related to the capacitance of the $C_f$.

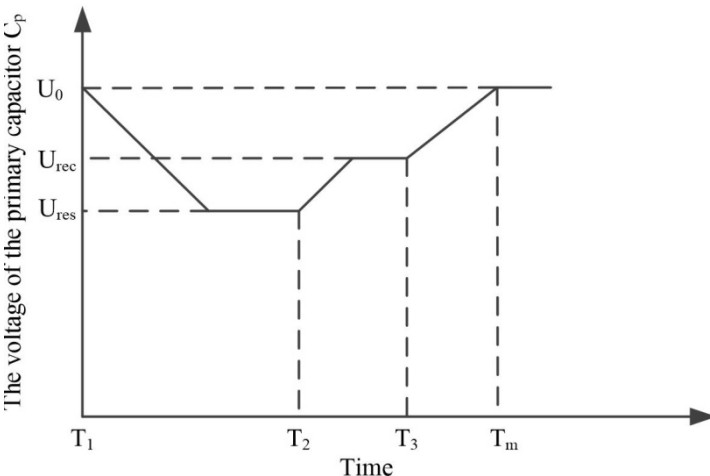

**Figure 10.** The sequence diagram of the thyristors.

### 3.2. Trigger Source

An important way of improving the synchronization performance of the switch is to reduce the rise-time of the trigger pulse and increase the trigger pulse amplitude. Here, a compact Marx generator, which is based on a magnetic switch, is designed for a trigger source [16]. The key point of this trigger source is that the integrated design of the transformer and the switch is realized, so that the volume of it can be reduced significantly. The basic circuit of the trigger source is shown in Figure 11. The magnetic switch is operated as a transformer during the charging process at first. When the magnetic core is saturated, the inductance of the secondary windings is decreased rapidly. At this time, the energy is sharply delivered to the load.

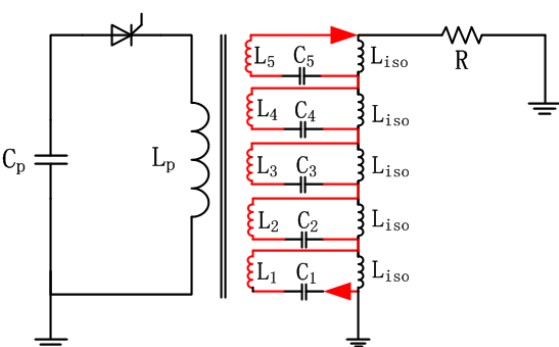

**Figure 11.** The circuit of the trigger source.

The trigger source assembly in the experiment is shown in Figure 12. It can be seen that the trigger source is very compact and its external dimensions are 150 mm × 200 mm × 50 mm. The Marx generator is assembled in a dielectric material house and pressurized $SF_6$ gas is used for electrical insulation. The essential characteristic of the magnetic switch is that the time of core saturation is consistent if the charging voltage of the primary capacitor is consistent. Therefore, the consistency of the trigger pulses is pretty good, as shown in Figure 13. The trigger source is operated in 30 Hz mode and the experimental results show that the amplitude of the trigger pulses is over 70 kV and the jitter is less than 10 ns. This trigger source provides a stable and reliable trigger pulse for the switches.

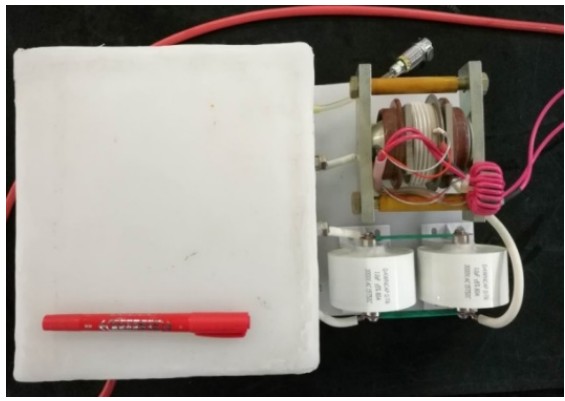

**Figure 12.** The photograph of the trigger source.

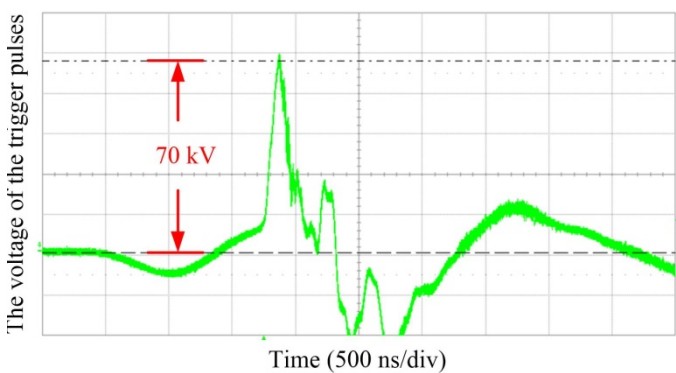

**Figure 13.** Screen shot of 20 pulses at 30 Hz operation of the trigger source.

### 3.3. Experimental Setup

In order to evaluate the performance of the compact modular PFN-Marx generator, the system of the generator was arranged as shown in Figure 14. The generator mainly consists of five parts, including a control subsystem, a trigger source, a primary energy subsystem, a PFN-Marx body, and a load. The charging voltage was measured with a commercial polestar probe with 2000:1. A water resister divider was inserted into the diode insulator to measure the load voltage. It can be seen in Figure 14 that the compactness assembly is accomplished. After the system is assembled in practice, the length of the PFN-Marx body is about 80 cm and its weight is about 56 kg.

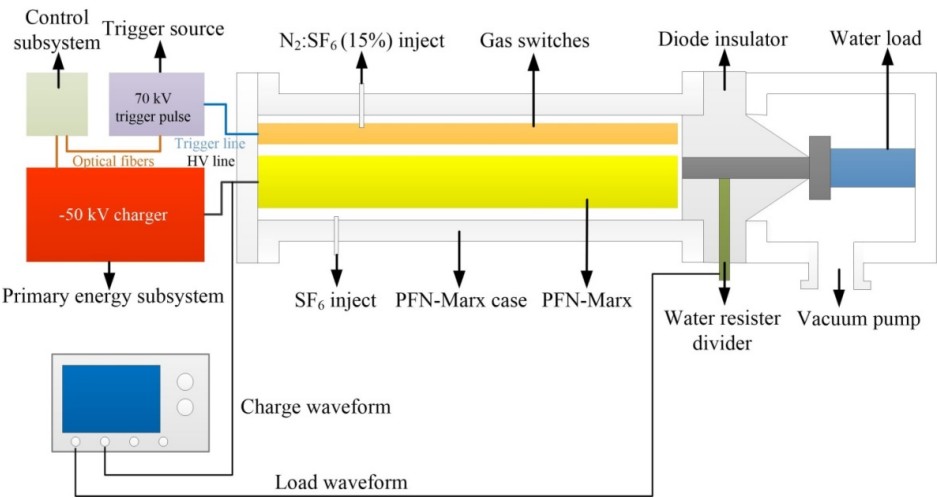

**Figure 14.** Schematic view of the PFN-Marx system.

## 4. Experimental Results

### 4.1. Experimental Results on a Water Load

Initial testing of the compact PFN-Marx generator involved firing the system at the charging voltage of 50 kV into a shot, and a 59 $\Omega$ coaxial $CuSO_4$ water load was connected to the generator to evaluate the operating characteristic of the generator. The water load was assembled in the vacuum diode as shown in Figure 14.

The generator was tested in single shot and burst mode, respectively. The pure $SF_6$ of pressure of 180 kPa was injected into the Marx casing for high voltage isolation, and the gas mixture of the $N_2$ and $SF_6$ with the pressure of 90 kPa was sent to the switch case. In single mode, a maximum 540 kV high-voltage pulse was delivered to the water load with the rise-time of 28 ns and the pulse width of 95 ns, as shown in Figure 15. The peak power of the output pulse was calculated to be about 5 GW ($P = U^2/R$) and the E/V and P/W of the PFN-Marx generator body were about 6.5 J/L and 90 MW/kg, respectively. The generator was also tested in burst mode. The load voltage pulse-train of the generator on the water load, which comprises five pulses at a repetition rate of 30 Hz, is shown in Figure 16. The charging voltage of the PFN-Marx is 45 kV and the amplitude of the load voltage is about 500 kV, which was measured via a resistance voltage divider. The dispersion of the load voltage is less than 3%. The load voltage waveform indicates that the operation of the generator in single mode agrees well with that in burst mode.

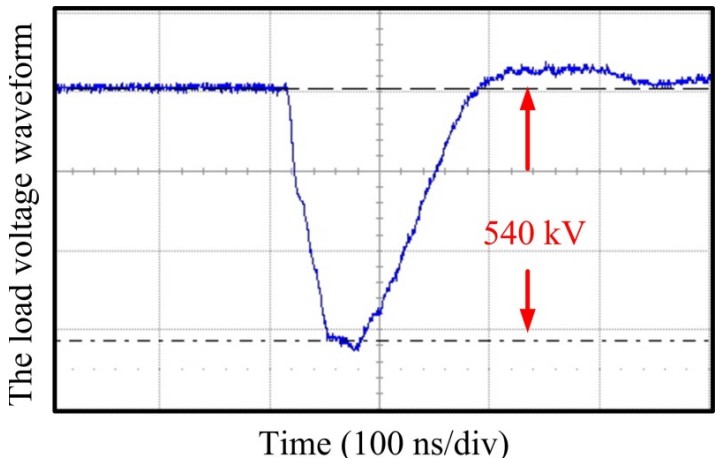

**Figure 15.** The load voltage waveforms on the water load.

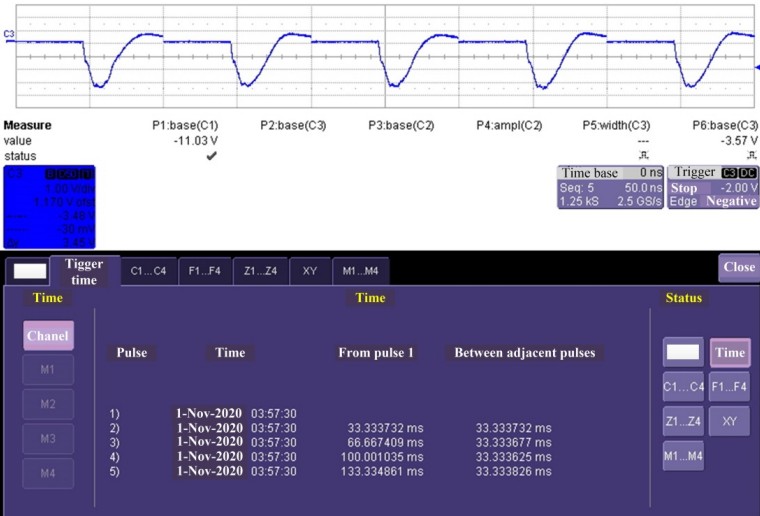

**Figure 16.** Screen shot of 5 pulses at 30 Hz operation.

However, the rectangular characteristic of the load voltage waveform in the experiment was not as good as in the simulation. The reason for that contains five parts. Firstly, the generator is constructed by multiple PFN modules, which are like building blocks. So, the coupling capacitor ($C_c$) between the adjacent PFN modules inevitably exists between the PFN modules, and the $C_c$, together with the PFN node inductors, forms a parasitic transmission line (PTL) with a short characteristic time. The final performance of the effect of the PTL is that the high-frequency oscillation is superimposed on the load voltage waveform. This is the main reason why the load voltage waveform's quality in the experiment was lower than that in the simulation. Secondly, the capacitor in the simulation is an ideal model and the mica in the practical experiment has self-inductance. In further work, the measure method of the mica capacitor's self-inductance will be studied. Thirdly, the inductance of the water resistor also affects the quality of the load waveform. Fourth, the load voltage was measured by a water resister divider with a low impedance type. The sample resistance of the resistor divider is 1 $\Omega$. The deterioration of the rise-edge of the load voltage waveform is significantly affected by the length of the connecting line of the resistance divider, which is the inductance of the connecting line ($L_m$). The dying oscillation of the load voltage waveform is decreased as the $L_m$ decreases. Finally, in the simulation analysis, the capacitance of the switch and the ground capacitance are ignored. These parasitic parameters also affect the load pulse. Actually, an oscillation damping circuit, which is to add an R-C unit between the last two PFN modules, has been preliminarily investigated in an experiment in another PFN-Marx system [17]. So, future work will also focus on the improvement of the load waveform's quality.

### 4.2. Experimental Results on a TTO

A miniaturized transit time oscillator (TTO) driven by this generator was initially investigated in an experiment. The setup for TTO operations is similar to the water load testing, with the dummy load removed. The Marx casing was filled with 160 kPa of pure $SF_6$, and, depending on the charging voltage, 80 kPa gas of $N_2$ mixed with 15% $SF_6$ was sent to the enclosed switch column. The generator was operated in triggering mode, and the output voltage and output current of the generator were measured by a resistance divider and a hand-made Rogovski coil, respectively. The typical waveforms of the generator and the microwave are shown in Figure 17. The pulse characteristics of the voltage waveform, including the voltage of 450 kV, the rise-time of 25 ns, and the pulse width of 90 ns, were measured in the oscilloscope. Obviously, the pulse width of the current waveform is shorter than that of the voltage waveform. The reason could be that the Rogovski coil has the defect during hand making. The sampling loop was not controlled to a minimum and the sampling resistor is not an inductive resistor. Besides, because the Rogovski coil is compact, there may be breakdown inside it during the measuring process. Therefore, distortion of the current signal exists. During the experiment, the Rogovski coil was used only for monitoring the electron beam production. A microwave pulse with power of about 100 MW and a pulse duration of 25 ns is generated when the diode voltage is 450 kV. The HPM signal was estimated to have a peak of 100 MW. The initial experimental results show the ability of the generator to drive the HPM source.

The TTO driven by the PFN-Marx generator in burst mode was also investigated initially in an experiment. The experimental results show that the TTO at 30 Hz with five pulses works well. The microwave power is about 85 MW, with the diode voltage about 440 kV, as shown in Figure 18. Obviously, there are some differences in the consistency of the microwave pulses. The experimental test with the PFN-Marx generator driving the TTO is still under study and detailed results will be published in the next work.

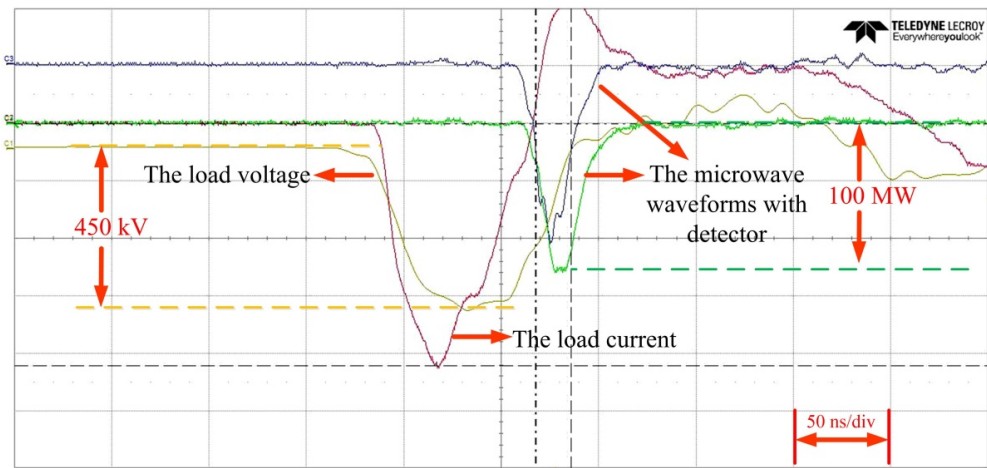

**Figure 17.** The typical waveforms on the transit time oscillator (TTO).

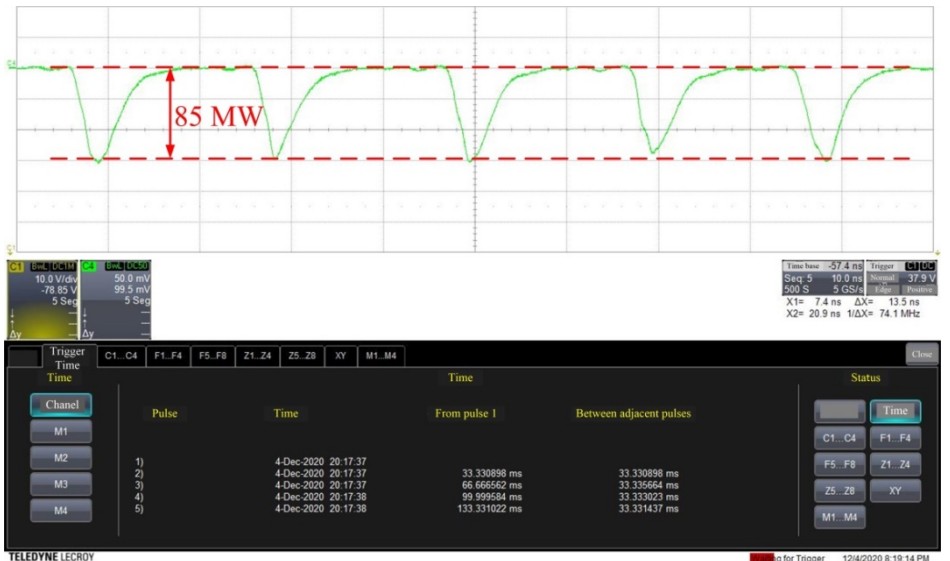

**Figure 18.** Screen shot of 5 microwave pulses at 30 Hz operation.

## 5. Summary

The 22-stage compact PFN-Marx generator based on the mica capacitors has shown successful 5 GW output on a water load. The performance of this PFN-Marx generator in burst mode is also described and the load voltage waveform indicates that the operation of the generator in single mode agrees well with that in burst mode. Typical pulse characteristics include a pulse width of 95 ns and a rise-time of 28 ns. A TTO HPM source driven by this generator was also tested and a 100 MW, 25 ns microwave pulse was obtained in an experiment.

A series of methods for compact and modular design of the PFN-Marx generator were used, including the large energy density mica capacitors, a mini-Marx trigger source with the integration of the magnetic transformer and switch, the PFN module design, a common switch case, and the connection between the PFN module and the switch. Under the condition of these designs, a single PFN module can be taken out from the PFN-Marx individually. Finally, the length of the PFN-Marx generator body is limited to 80 cm and its weight is about 56 kg. The ratio of the energy storage to volume and the ratio of power to weight of the 22-stage PFN-Marx generator are up to 6.5 J/L and 90 MW/kg, respectively.

Actually, a lot of work could be done to improve the performance of the generator; for example, increasing the pulse number and improving the quality of the output waveform,

as well as optimizing the operation of the TTO source driven by the generator. Furthermore, based on our generator, other types of HPM sources, including a relativistic magnetron and a relativistic backward wave oscillator, are being studied.

**Author Contributions:** PFN-Marx design and structure design, H.Z. (Haoran Zhang); validation, T.S.; primary energy subsystem, S.L.; methodology, Z.Z.; TTO design, L.S.; control subsystem, H.Z. (Heng Zhang); writing—original draft preparation, H.Z. (Haoran Zhang); writing—review and editing, T.S. and Z.Z. All authors have read and agreed to the published version of the manuscript.

**Funding:** This research received no external funding.

**Institutional Review Board Statement:** Not applicable.

**Informed Consent Statement:** Not applicable.

**Data Availability Statement:** The data that support the findings of this study are available from the corresponding author upon reasonable request.

**Acknowledgments:** The authors wish to thank Bo Liang and Jia Li for their support during the assembly and the experiment.

**Conflicts of Interest:** The authors declare no conflict of interest.

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
