# Peer review of "A Compact Modular 5 GW Pulse PFN-Marx Generator for Driving HPM Source"

_electronics, doi:10.3390/electronics10050545_

Round 1

Reviewer 1 Report

Thanks for this interesting research. It is a worthy subject and the developed platform looks attractive;
However I believe it is worth developing the paper further. It is in an early phase and needs a lot of additions, refinements, and clarifications. Therefore, although a valuable contribution, the paper requires a very thorough revision.  
1.  You are requested to define what is your true objective?
2. The quality of figure must be improved.
3. I kindly invite you to add a result 

Reviewer 2 Report

The paper contains abundantly the main elements specific to the field of GW pulse PFN-Marx generators. A minor remark related to the use of the word photograph (row 100) that is not in perfect agreement with what is found in the Figure 3 a synonymous word frequently used in the field of electronics is recommended. In conclusion, the paper is a real contribution to the technology of pulse PFN-Marx generators.

Reviewer 3 Report

The article reports on the design, assembly and experimental testing of a PFN Marx generator. The results demonstrated essentially complement and even overcome the ones authors are referencing to. Nevertheless, there are certain issues that authors still need to elaborate on for the article to be considered decent of publication.

From the introduction, it is not quite clear what are the fundamental issues that have been actually resolved by the authors with respect to the listed solutions. Moreover, the narrative at this point is perceived as rather superficial. Authors are suggested to pay more attention to the details that basically were used as the starting motivation for the work. In this way, only the reader will understand better the depth of the authors’ contribution.

The text, in general, requires detailed proofreading, paying special attention to grammatical issues. Otherwise, it significantly deteriorates the perception of the narrative. Authors are should pay more attention to the figures editing, because in this way they are unacceptable (as being organized in tables, see Figures 1 and 2).  Please provide more details to the Figures 3 and 4, e.g. geometric sizes, specification of materials, some spatial references (top, bottom view), try to transfer this information from the text to the figures (contained in the lines 131-146). It will make significantly simplify the text perception. For Figure 13 please provide the ordinate axis values. Figure 16 (p. 9) reports the incorrect date of measurements, please consider it, otherwise, an attentive reader may suspect that the screenshots were taken from the work performed at least 17 years ago. Also, with all the respect to the Chinese language, it would be much more appropriate to have the column titles in English.

Below the list of several omissions is presented, please take into account that it is by no means complete. And the issues provided are meant to be considered as examples.

Line #

Comment

86

Pay attention to the grammar: eliminate the repeated “the” and put “-ed”  after “list”

100

The image in figure 3 doesn’t seem to be a photo, rather CAD design, please correct.

105

Given in this way (without an article), “compactness” seems to be an adjective, which, I believe, is not the case. Please correct.

110

Check, whether the preposition (probably “in”) is omitted between “adopted” and “which”.

121

Eliminate “of the switch”

122

Inasmuch Figure 1 is a schematic diagram, to be used as a reference for the physical localization of the switches doesn’t make sense. It would be more reasonable to provide a CAD image.

164

Leave just “assembled”, but not “assembled them”

188

Correct “An important”

239-240

Correct “inevitably existed”

247

Correct “resistor”

234-255

Within all the possible reasons for the rectangular characteristic of the load's voltage to be far from the simulated one, the inherent non-linearity of thyristors has not been specified. Given that the problem may partially be considered as transient phenomena suppression, I would recommend you to have a look at RC snubber networks. Consider, that this particular aspect may contribute essentially as well.

266-267

Consider rephrasing of the sentence on Rogovski coil defect. In actual form, it’s uneasy to interpret.

p. 10, Figure 17

Correct the number under the figure.  Provide a clear y-axis label as well.

266-288

Please take into account that the values 6.5 J/L and 90 MW/kg cannot be representative when considered in respect to a single PFN module, given that in total you use 22 of them. Therefore, it makes it impossible to compare the final results with the referenced ones, unless you mean that these values are cumulative, i.e. regarding the whole PFN-Marx generator. The issue is quite crucial since every new work is supposed to overcome the things reported previously.

Round 2

Reviewer 3 Report

The article has been essentially improved. The Reviewer remained totally satisfied with the content of the Cover Letter provided. Therefore, the article may be considered for publication. 

Congratulations to the authors for the work performed.